**Data Availability Statement:** The raw data supporting the conclusions of this article have been deposited in the NCBI database (https://www.ncbi.

# Lower respiratory tract microbiota in patients with clinically suspected nontuberculous mycobacterial pulmonary disease according to the presence of gastroesophageal reflux

Eun Jeong Won[1]☯*, Yu Jeong Lee[2,3], Moon-Ju Kim[3], Tae-Jong Kim[3], Hong-Joon Shin[4], Tae-Ok Kim[4], Yong-Soo Kwon[4]☯*

1 Department of Laboratory Medicine, Asan Medical Center, University of Ulsan College of Medicine, Seoul, Republic of Korea, 2 Department of Biomedical Sciences, Graduate School of Chonnam National University, Gwangju, Republic of Korea, 3 Department of Rheumatology, Chonnam National University Hospital, Gwangju, Republic of Korea, 4 Department of Internal Medicine, Chonnam National University Hospital, Chonnam National University Medical School, Gwangju, Republic of Korea

☯ These authors contributed equally to this work.
* yskwon@jnu.ac.kr (Y-SK); ejwon@amc.seoul.kr (EJW)

## Abstract

Although gastroesophageal reflux has been recognized as one of the risk factors of nontuberculous mycobacterial pulmonary disease (NTM-PD) progression, the effect of reflux on the lower respiratory tract microbiota has not been studied in detail. We investigated the composition of the lower respiratory tract microbiota in patients with clinically suspected NTM-PD, comparing them based on the presence of reflux. Forty-seven patients suspected of having NTM-PD were enrolled and assigned according to presence of reflux (n = 22) and non- reflux (n = 25). We performed a pepsin ELISA assay to identify the presence of reflux and 16S ribosomal RNA gene amplicon sequencing to evaluate the microbiota in bronchoalveolar lavage fluid. There were no significant differences in the diversity or composition of the lower respiratory microbiota between the NTM-PD and non-NTM-PD groups. Bacterial richness was observed in the non-reflux group than in the reflux group [$P = 0.03$] and a cluster in the reflux group was observed. The reflux group showed a predominance for *Pseudomonas aeruginosa* or *Staphylococcus aureus* among the NTM-PD group and for *P. aeruginosa*, *Haemophilus influenzae*, *Klebsiella pneumoniae*, or *Eikenella* species among the non-NTM-PD group. The non-reflux groups presented diverse patterns. A linear discriminant analysis and volcano plot demonstrated that *P. aeruginosa*, *H. haemolyticus*, *Selenomonas artemidis*, and *Dolosigranulum pigrum* were specifically associated with the NTM-PD reflux group, while *P. aeruginosa* was specifically associated with the non-NTM-PD reflux group. These observations confirm that the lower respiratory microbiota is consistently altered by reflux but not in NTM-PD.

[nlm.nih.gov/bioproject/PRJNA981296](nlm.nih.gov/bioproject/PRJNA981296)) and will be made available by the authors upon request.

**Funding:** This study was supported by grants from the National Research Foundation of Korea (NRF) Grant funded by the Ministry of Education, Science, and Technology (grant no. NRF-2022R1C1C1002741 (Eun Jeong Won) and 2020R1F1A1076570 (Yong-Soo Kwon)). The funding body played no role in the design of the study and collection, analysis, interpretation of data, and in writing the manuscript.

**Competing interests:** The authors have declared that no competing interests exist.

## Introduction

Nontuberculous mycobacterial pulmonary disease (NTM-PD) is a growing chronic health concern that is challenging to treat and has been increasing in incidence worldwide [1]. Prolonged anti mycobacterial chemotherapy are frequently required in order to manage the disease effectively. Previous studies have found that NTM-PD is associated with comorbidities such as chronic obstructive pulmonary diseases, osteoporosis, gastroesophageal reflux (GERD), and cystic fibrosis [1]. It has been reported that up to 45% of individuals diagnosed with NTM-PD also experienced GERD, linked to more severe symptoms and disease [2,3]. In addition, a recent study reported that patients with GERD had about a 3.4-fold higher risk of developing NTM-PD than those without GERD [4].

The precise mechanism by which GERD conditions increase the likelihood of NTM-PD is not yet fully understood. However, it is suggested that micro-aspiration resulting from GERD could be one potential means by which the microorganisms is able to migrate to the lungs [5]. Previously, it was widely believed that the lungs of healthy individuals were sterile, as culture-positive samples were absent. Recently, however, several studies have demonstrated the presence of bacterial, fungal, and viral communities in the lungs of both healthy and diseased individuals [6]. Three primary factors have been proposed to influence the lung microbiome as follows: the extent to which microorganisms enter the lungs, the rate at which microorganisms are eliminated from the lungs, and the growth of the microorganisms comprising the microbiome [7]. When investigating differences between health and disease, it is crucial to consider the contribution and equilibrium of these factors to the respiratory microbiome. Although the causal relationships between the dynamics of the microbial community and clinical outcomes remain unclear, there are several observed correlations in patients with cystic fibrosis or bronchiectasis [8,9]. Notably, the progressive loss of diversity and subsequent microbial dysbiosis is evident in cystic fibrosis with an increase in the dominance of pro-inflammatory pathogens, a key factor underpinning disease severity [10]. Identifying the role of the microbiome in NTM-PD may provide insight into the management of the disease, but available studies are limited in this regard.

It is not always easy to diagnose NTM-PD patients promptly because, in many cases, sputum culture results were consistently negative or the patients were unable to produce adequate sputum. In these cases, clinicians performed bronchoscopy for detecting NTM in their lower respiratory tract. Here, we investigate the lung microbiota of the patients with suspected NTM-PD and further analyzed them regarding the impact of reflux.

## Materials and methods

### Patients and sample collection

Patients who underwent a bronchoscopy due to a suspicion of NTM-PD according to a chest CT in a 1000-bed tertiary university medical center in Gwangju, Republic of Korea between October 2020 and August 2022 were prospectively enrolled in the NTM-PD group after providing written informed consent. NTM-PD was suspected when the patients had typical findings of the disease on the chest CT, characterized by bilateral multilobar bronchiectasis and small nodules predominantly observed in the right middle lobe and the lingular segment of the left upper lobe, with symptoms that were consistent with those who have NTM-PD such as chronic cough and sputum [11]. We enrolled patients with clinically suspected NTM-PD, and only those with bacteriologically confirmed NTM-PD met the diagnostic criteria of the ATS/IDSA guidelines [12]. The exclusion criteria were malignancy at any site, use of any antibiotics during the previous month, perceived vulnerability, and refusal to participate. All the patients

underwent bronchoscopy with bronchoalveolar lavage (BAL). During bronchoscopy, BAL samples were collected by instilling 30 mL of 0.9% NaCl three times into the lung at a subsegmental level, where the lesions appeared to have the most active infection with micronodules and bronchiectasis. The first and third fractions of specimens were sent for a diagnostic microbiological evaluation as part of standard care. Second fraction of specimen was immediately transferred onto ice to lab. Samples were promptly spun at 10,000 g, within 30 minutes of BAL sample collection. The cell pellets in 1 mL were stored at −80˚C. The study protocol was reviewed and approved by the Chonnam National University Hospital Institutional Review Board (CNUH-2020-259). We received written informed consent from all the participants in this study.

## Levels of pepsin in the lower respiratory specimens

Levels of pepsin in the BAL samples were quantified using a sandwich ELISA kit [Wuhan Fine Biotech Co., Ltd, China]. Detecting pepsin in BAL fluid serves as evidence of micro-aspiration even in the absence of symptomatic reflux events [13]. However, it is crucial to note that the levels of pepsin may vary depending on the testing methods and the populations enrolled in studies. We have adopted cutoff levels based on a previous study showing that a cutoff of 100 ng/ml can effectively differentiate mean pepsin levels between patients with and without anti-acid treatment [14]. The study group was divided into a microaspiration group (with levels of pepsin ≥100 ng/mL) and a non- microaspiration group (with levels of pepsin <100 ng/mL), respectively.

## Microbiota analysis of the lower respiratory specimens

DNA was extracted from stored BAL pellets using a GeneAll Exgene Blood SV mini kit (Seoul, Korea), following the instructions provided in the manufacturer's manual (version 07272016). The V3-V4 regions of the bacterial 16S rRNA genes were amplified and were sequenced using the 2 × 300 paired-end MiSeq kit (Illumina, San Diego, CA, United States) by Macrogen (Seoul, Korea). The targeted sequences were analyzed as previously described [15,16]. Briefly, after removing low-depth samples (<9,000 sequences per sample), the sequences underwent trimming, merging, and clustering into operational taxonomic units (OTUs) using CLC Genomics Workbench v. 10.1.1 and CLC Microbial Genomics Module v. 2.5 (Qiagen, Hilden, Germany). Taxonomic classification of the sequences was performed by referencing the National Center for Biotechnology Information taxonomy database, with an OTU cutoff of 3%. The alpha diversity indicating richness and Shannon's index were calculated using rarefied OTU counts [17]. Exploratory analysis of the beta-diversity [between-sample diversity] was performed based on the Bray-Curtis measure of dissimilarity as a principal coordinate analysis. Hierarchical cluster analysis was performed based on Bray-Curtis metrics and complete linkage clustering. The linear discriminant analysis (LDA) effect size, available at http://huttenhower.sph.harvard.edu/galaxy/, was utilized to determine the magnitude of differential abundance for each taxon, employing criteria of LDA ≥ 3.0 and $P < 0.05$ [18]. Volcano plots were generated to depict the estimated log 2-fold differences in OTU abundance, categorized by reflux, within each subgroup. In addition, the microbial metabolic function according to the presence of reflux was predicted by PICRUSt2 and MetaCYC database [19].

## Statistical analysis

The non-parametric factorial Kruskal-Wallis sum rank test was used to test for differences in the two subgroups. The statistical significance of beta diversity was obtained by the Permutational multivariate analysis of variance test. Statistical analysis was performed using CLC Genomics Workbench v. 10.1.1 (Qiagen, Hilden, Germany) and the GraphPad Prism 9.0 software (GraphPad Software Inc., San Diego, CA, United States).

### Ethics statement

This study was carried out in accordance with all the relevant institutional guidelines. The Ethics Committee of Chonnam National University Hospital approved this study (CNUH-2019-134) and written informed consent was obtained from all the subjects.

## Results

### Characteristics of the patients with suspected NTM-PD

A total of 47 patients with suspected NTM-PD (22 with reflux and 25 without reflux) were included in this study. Of all patients, 24 patients were bacteriologically confirmed NTM-PD but 23 patients were not bacteriologically confirmed (non-NTM-PD). All patients were undergoing treatment for a naïve and nodular bronchiectatic form of the disease. No chest cavities were observed in patients according to the CT scan. Females were more common among the patients with reflux; however, there were no differences in other baseline characteristics (Table 1). There were no differences in baseline characteristics between patients with and without bacteriologically confirmed NTM-PD (S1 Table).

### Diversity in the lower respiratory microbiota according to reflux

During the two steps of library quality control, 23 samples were excluded because of a low quantity of DNA and finally 47 samples were included in this study. A total of 2,027,154 good-quality reads with a mean length of 301 base pairs were generated. There was no significant difference in the diversity of the lower respiratory microbiota between patients with and without bacteriologically confirmed NTM-PD. Overall, the reflux group showed a lower diversity of the lower respiratory microbiota than that of the non-reflux group ($P = 0.03$) (Fig 1). Notably, the reflux group among patients without bacteriologically confirmed-NTM-PD presented with a significantly lower diversity of the lower respiratory microbiota than the non-reflux group ($P = 0.01$). Contrarily, among the bacteriologically confirmed NTM-PD group, there was no significant difference in the diversity of the lower respiratory microbiota according to reflux. When we compared the overall composition of the lower respiratory microbiota (Fig 2), a cluster was noticed in the reflux group, distinctive from the dispersed non-reflux group. We could not find significant differences in the bacterial load, or in the α- or β-diversity of the bacteriologically confirmed versus non-bacteriologically confirmed group (S1 Fig).

**Table 1. Baseline characteristics of the patients with clinically suspected non-tuberculosis mycobacterium pulmonary disease (NTM-PD) enrolled in this study.**

| | Total (n = 47) | Patients with clinically suspected NTM-PD with reflux (n = 22) | Patients with clinically suspected NTM-PD without reflux (n = 25) | P-value |
|---|---|---|---|---|
| **Age, median (IQR)** | 63.0 (56.0–74.0) | 67.0 (57.8–75.5) | 63.0 (55.5–74.0) | 0.276 |
| **Sex (male), n (%)** | 17 (36.2) | 3 (13.6) | 14 (56.0) | 0.005 |
| **Ever smoking, n (%)** | 11 (23.4) | | | |
| **Previous tuberculosis, n (%)** | 7 (14.9) | 4 (16.7) | 3 (13.0) | 1.000 |
| **Diabetes, n (%)** | 4 (8.5) | 2 (9.1) | 2 (8.0) | 1.000 |
| **Heart disease, n (%)** | 5 (10.6) | 2 (9.1) | 3 (12.0) | 0.666 |
| **Chronic kidney disease, n (%)** | 2 (4.3) | 2 (9.1) | 0 (0) | 0.214 |

Abbreviations: IQR, Interquartile range.

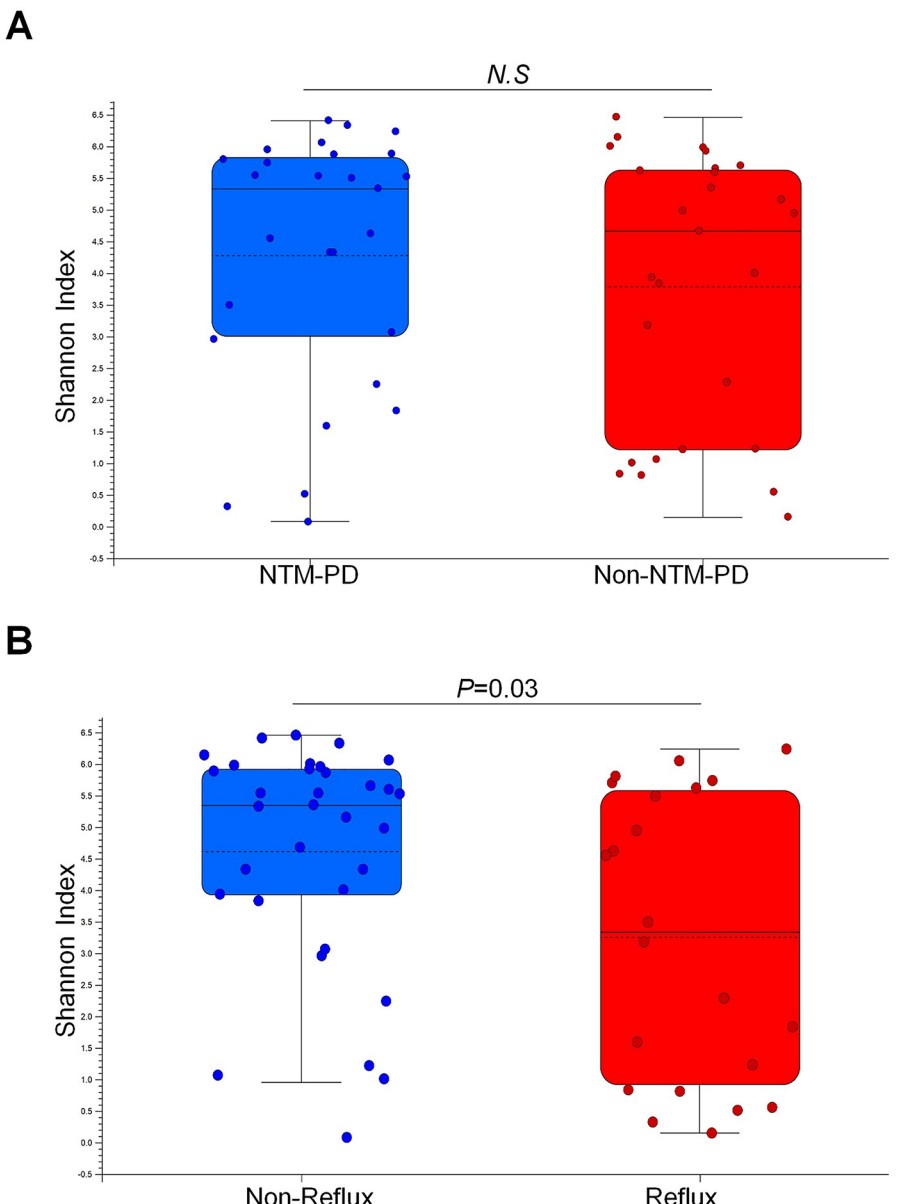

**Fig 1. Alpha diversity indices (Shannon index) of patients with clinically suspected nontuberculous mycobacterial pulmonary disease (NTM-PD) according to the reflux.** The Shannon index of the bacteriologically confirmed NTM-PD group was not significantly different from that of the non-bacteriologically confirmed NTM-PD group (non-NTM-PD) ($P = 0.4$)(A). The Shannon index of the reflux group was significantly lower than that of the non-reflux group ($P = 0.03$)(B). Among the bacteriologically confirmed NTM-PD group, the Shannon index was not significantly different according to the reflux ($P = 0.4$)(C). Among the non-NTM-PD group, the Shannon index of the reflux group was significantly lower than that of the non-reflux group ($P = 0.01$)(D).

## Bacterial composition of the lower respiratory microbiota at the species level

Thereafter, we compared the distribution of the lower respiratory microbiota at the species level, according to reflux (Fig 3). Among the bacteriologically confirmed NTM-PD group, the reflux group showed a predominance of *Pseudomonas aeruginosa* or *Staphylococcus aureus*,

**A**

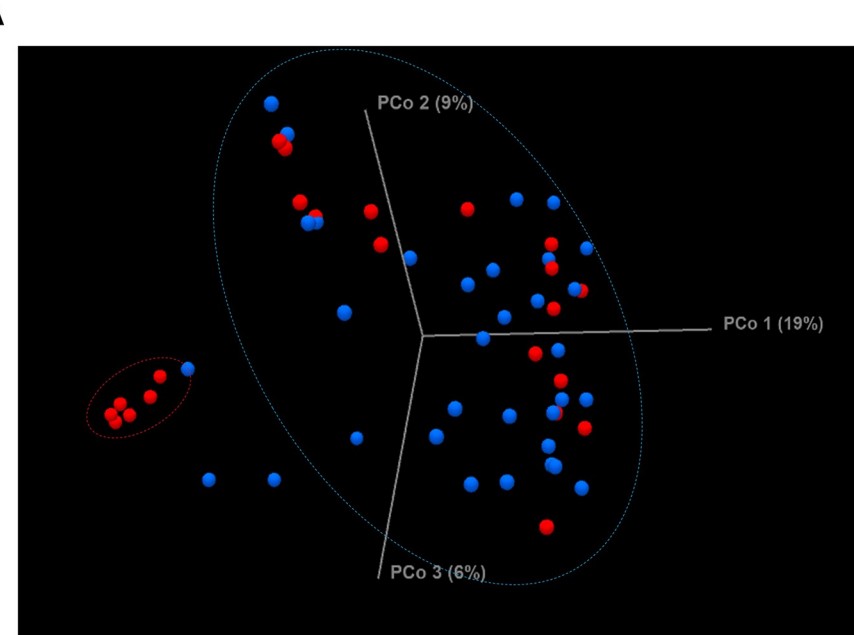

**B**

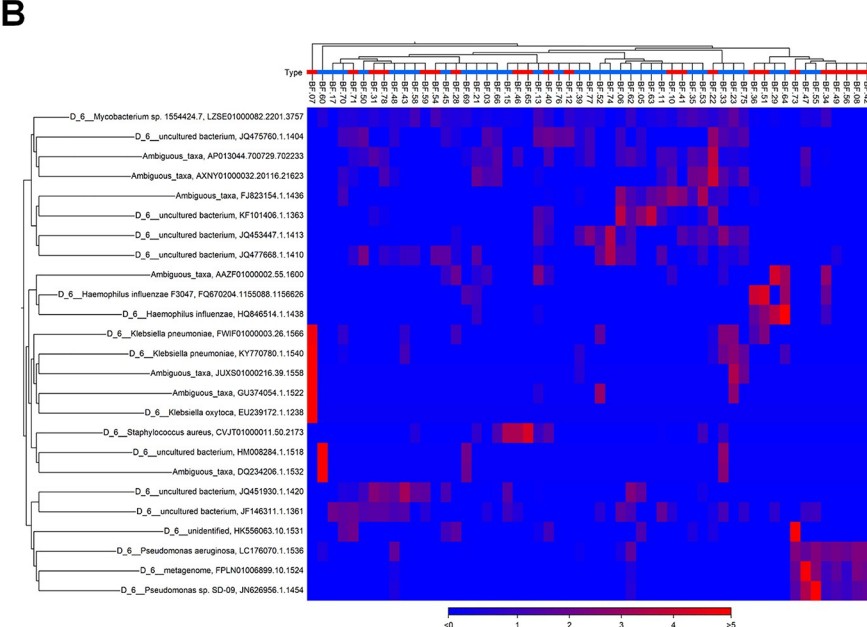

**Fig 2. Composition of the lower lung microbiome in clinically suspected NTM-PD patients with reflux.** A distinct cluster of the reflux group (red) was noticed apart from the dispersed non-reflux group (blue)(A). Unweighted beta diversity analysis showed the overall bacterial community structure and phylogenetic diversity of the reflux group (red) and non-reflux group (blue)(B).

indicating that dysbiosis of the lung microbiota is closely related to the existence of reflux. Among the non-NTM-PD group, the reflux group showed a predominance of *P. aeruginosa*, *Haemophilus influenzae*, *Klebsiella pneumoniae*, or *Eikenella* species. In contrast, the non-reflux groups presented diverse patterns without any predominance of pathogens, in either the

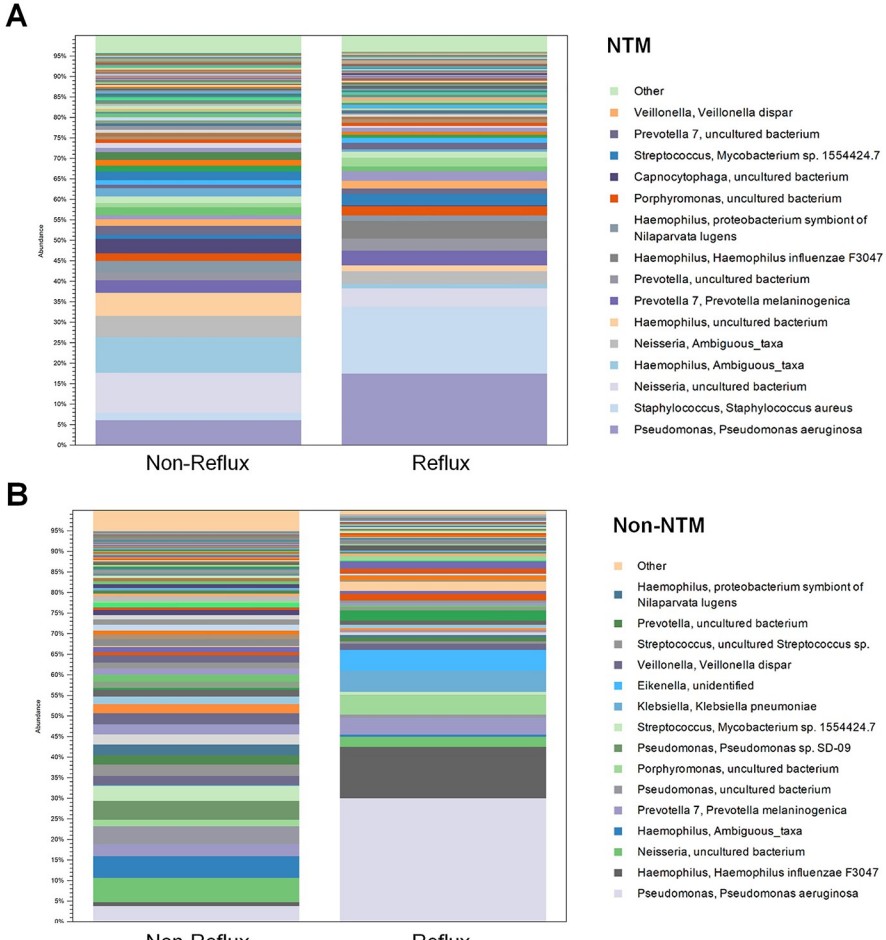

**Fig 3. Composition of the lower lung microbiome at the species level in the bacteriologically confirmed NTM-PD group (A) and non-NTM-PD group (B) with reflux.**

bacteriologically confirmed NTM-PD or not. Each taxonomical level was compared using LDA to determine the bacterial taxa associated with reflux or NTM-PD (Fig 4). Overall, *S. aureus* was highly correlated the bacteriologically confirmed NTM-PD group but *K. pneumoniae* was highly correlated with the non-NTM-PD group. *Prevotella* 1 was highly correlated with the reflux group but several taxa such as *Haemophilus*, *Neisseria*, *Prevoella 7*, *Flavobacteriaceae*, *Capnocytophaga*, *Alloprevotella*, *Veillonella*, *Leptotrichia*, *Fusobacterium periodonticum*, *and Neisseria meningitidis*, were found to be highly correlated with the non-reflux group. Within the non-NTM-PD group, *Pseudomonas* species were highly correlated with reflux group, while several taxa were found to be related to the non-reflux group. Within the bacteriologically confirmed NTM-PD group, only *Lautropia* was specific to the non-reflux group (S2 Fig). Volcano plots also showed that *P. aeruginosa*, *H. haemolyticus*, *Selenomonas artemidis*, and *Dolosigranulum pigrum* were specifically associated with the bacteriologically confirmed NTM-PD reflux group, while *P. aeruginosa* was specifically associated with the non-NTM-PD reflux group (Fig 5). Furthermore, to determine any functional enrichment or depletion of lower respiratory tract microbiota in relation to reflux, functional assessment through PICRUSt2 was performed to analyze the KEGG pathway compositions in microbial

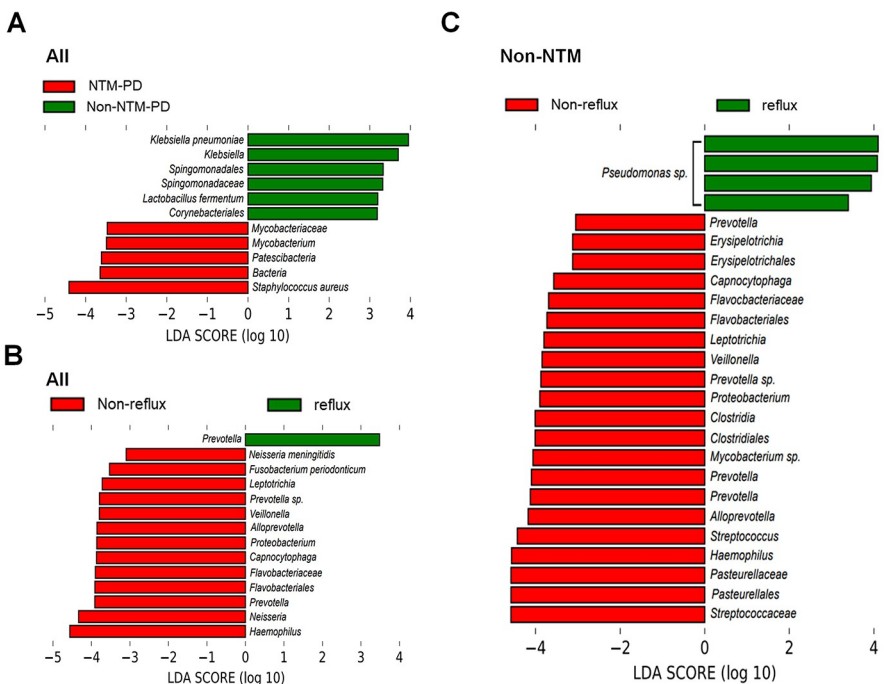

**Fig 4. Specific bacterial taxa are associated with reflux among patients with clinically suspected NTM-PD, according to the linear discriminant analysis effect size analysis.** (A) Bacteriologically confirmed nontuberculous mycobacterial pulmonary disease (NTM-PD) vs. non-NTM-PD group (B) Reflux vs. non-reflux group (C) Reflux vs. non-reflux among the non-NTM-PD group. The length denotes the effect size of a taxon. $P = 0.05$ for the Kruskal–Wallis test; the LDA score is >3.0.

populations. Several functional categories related to aerobactin biosynthesis, nucleic acid/protein/iron metabolism, energy production, or other cellular metabolism were up-regulated in reflux group (S3A Fig). To the contrary, categories related to antioxidant defense or production of steroid hormones were down-regulated, as such lysine metabolism and steroid biosynthesis in reflux group (S3B Fig).

## Discussion

It is essential to understand the signature of the lung microbiome niche of chronic lung diseases such as NTM regarding the predisposition of GERD. We notice that the lower respiratory microbiota is consistently altered by reflux, and a predominance of *P. aeruginosa* was noticeable to be associated with reflux, irrespective of NTM-PD or not. Previously, the lower airways of healthy individuals have been shown to harbor diverse low-abundance bacterial communities, while the lower airways in patients with respiratory disease consist of bacterial communities with shallow diversity and high abundance [20,21]. Previously thought to be a sterile environment, lungs normally harbor a diverse microbiome of low microbial biomass that is composed of *Prevotella*, *Streptococcus*, *Veillonella*, *Fusobacterium*, and *Haemophilus* genera [22,23]. We observed that the common components of the lung microbiota mentioned above were found to be highly correlated with the non-reflux group. Although these observations may not be broadly applicable to all subgroups of patients, the results of the study indicate that the niche of the lung microbiome without reflux may be healthier than the condition with reflux. We could not find significant differences in the bacterial load, or in the α- or β-

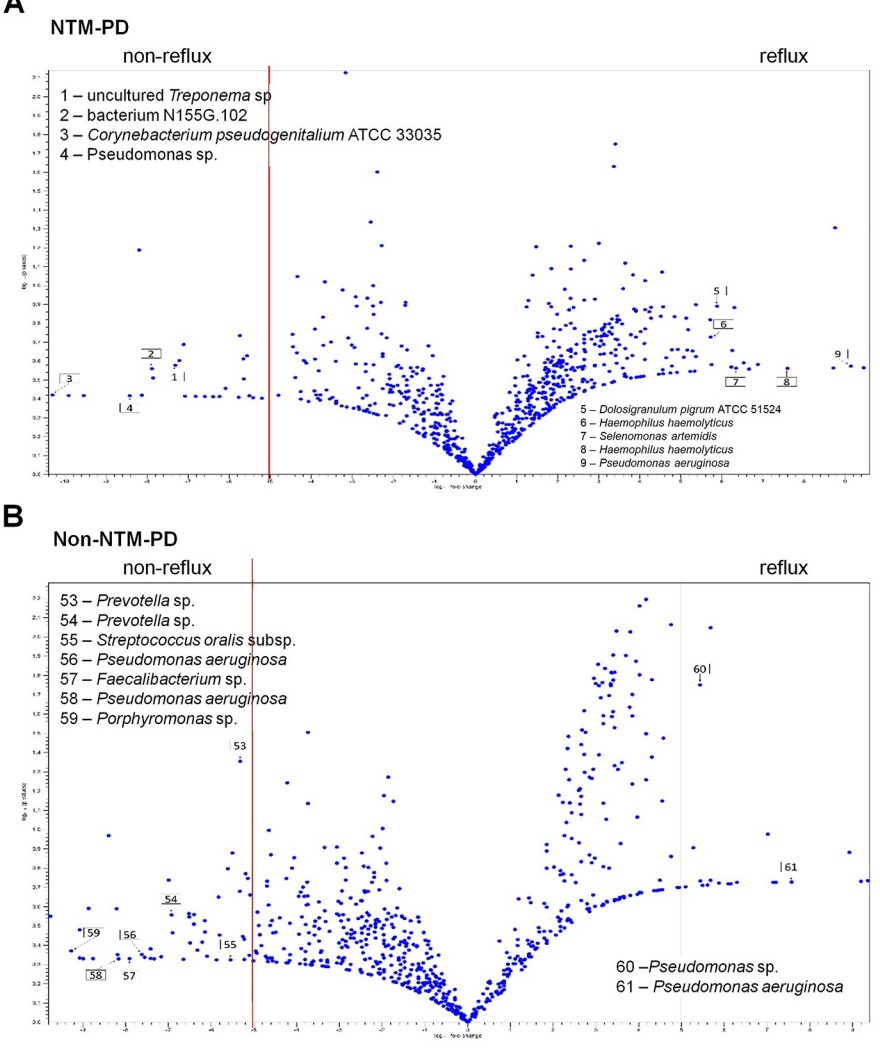

**Fig 5. Volcano plot showing several bacterial taxa specifically related to reflux among patients with clinically suspected NTM-PD.** Bacteriologically confirmed nontuberculous mycobacterial pulmonary disease (NTM-PD) (A) and non-NTM-PD (B).

diversity of the bacteriologically confirmed NTM-PD versus non-NTM-PD group, and this was partly in line with previous data published by Sulaiman et al. [24]. Several confounding factors included in the heterogenetic NTM disease may make it difficult to apply general conclusions about the microbial communities across diverse diseases.

Yamasaki et al. firstly performed an NTM microbiome study on BAL samples from patients with NTM-PD [25]. In that study, lower rates of *Haemophilus*, *Pseudomonas* and *Staphylococcus* abundance, but a higher rate of *Streptococcus* was observed in patients with NTM-PD compared to the controls. Anaerobic species, primarily *Prevotella*, *Fusobacterium*, *Propionibacterium*, and *Veillonella*, were significantly more abundant in patients with NTM-PD, suggesting the potential role of anaerobes attributing mycobacterial growth-inducing local tissue hypoxia. In our dataset, we observed a strong correlation between *Prevotella* and the reflux group, indicating that a significant proportion of them likely originated from

the gut, in addition to baseline of lung commensals [26]. *Fusobacterium periodonticum* was noticed as one of the marker of non-reflux group, potentially originating from oral cavity [27]. These findings might be partly explained by underlying lung disease condition. Basically, the gut is anaerobic, the lungs are aerobic during health. However, during disease condition, air-flow markedly changes in the lower bronchi and bronchioles due to various structural and inflammatory factors. These changes can significantly impact the migration of microbes into the airways. In addition, conditions such as reflux can markedly affect the dynamics of microbial elimination from the airways via impairment of ciliary function, mucus secretion increase, and mucus viscosity alteration. This can also lead to changes in airway temperature and further create anaerobic zones for microbial growth in the lungs. The ongoing inflammatory processes, which create anaerobic zones, cause serum leak or mucus accumulation in the alveoli, or result in alveolar collapse can markedly change the microbiome of the lungs, which in turn, has the potential to further drive inflammatory processes [28]. Above all, we found a predominance of *P. aeruginosa* or *S. aureus* in lower lung microbiome suggesting the dysbiosis of the lung microbiome, closely related to the existence of reflux. This finding was supported by the notion that chronic infections with *P. aeruginosa* have been shown to drive this pulmonary dysbiosis, rather than the existence of prior dysbiosis facilitating the emergence of the pathogen [29]. While causative relationships between the dynamics of the microbial community and the clinical outcomes are unclear, several correlations have been observed [9,30]. Several *in vitro* studies have shown that *P. aeruginosa* inhibits the growth of *H. influenzae*, suggesting competition within the lung ecological niche [30]. From a clinical perspective, this observation suggests an antagonistic relationship between these pathogens, indicating their inability to be part of a single core microbiome. The relevance of a "core" taxa [dominated by *P. aeruginosa*] and the loss of richness in the microbiome, indicating worse lung function, has been previously aroused [19]. In addition to the impact of chronic *P. aeruginosa* colonization on the structure of the lung microbiome, other factors, such as antibiotics, environmental cues, architectural distortion, local immune tone or systemic immune status, could played a role in the shaping of the lung microbiota [31–33]. Taken together, we ensured the implication of *P. aeruginosa* colonization on the lung microbiome which was closely related to reflux, rather than NTM-PD. This notion highlights the importance of the diagnosis and management of reflux in patients with chronic lung disease, including NTM-PD. Further study would be needed to clarify how pepsin levels are associated with lower airway dysbiosis, and how they impact longitudinal changes in the lower microbiota. However, *P. aeruginosa* is commonly associated with structural lung diseases such as bronchiectasis, potentially leading to lung parenchymal damage [34]. Therefore, patients with reflux may exhibit more severe lung involvement and a higher prevalence of *P. aeruginosa* colonization. Our analysis categorized the extent of lung involvement into mild, moderate, and severe based on the number of affected lobes and the presence of bilateral disease or cavities in chest CT scans. Among our patient cohort, 25 individuals (53.2%) exhibited mild disease, while 22 (26.8%) showed moderate disease; no severe cases were recorded. Notably, patients with reflux were more likely to present with moderate disease compared to those without reflux [17 (77.3%) vs. 5 (20.0%), $P = 0.001$], indicating a greater extent of lung involvement in this subgroup. However, it remains unclear whether this difference in disease severity correlates with alterations in the microbiome, posing a potential limitation to our study.

Our functional prediction data indicated that reflux itself might impede the maintenance of homeostasis and cell protection during oxidative and osmotic stresses within bacterial communities. Of note, reflux led to the up-regulation of aerobactin biosynthesis and various pathways crucial for cellular metabolism. The excessive production of aerobactin has been suggested as a key factor in the heightened virulence of the invasive form of *K. pneumonia*

[35]. These findings suggest that reflux may modulate specific microbial niches, making them more susceptible to pathogenic enteric Gram-negative bacteria.

Unfortunately, our study faced additional several limitations as below. Firstly, the small sample size lacked adequate statistical power and we could not obtain any samples recovered from bronchoscopy not associated with NTM-PD or healthy conditions. In terms of study cohort, we enrolled patients with clinically suspected NTM-PD, and only those with bacteriologically confirmed NTM-PD met the diagnostic criteria of the ATS/IDSA guidelines [12]. Five of the non-bacteriologically confirmed NTM-PD cases revealed *M. tuberculosis* in their final BAL specimen culture results. The patients without bacteriological confirmation of NTM-PD may have different underlying diseases. Further, we did not know the exact underlying causes of bronchiectasis in the enrolled patients, although bacteriologically confirmed NTM-PD could have bronchiectasis due to NTM. Secondly, we designated the reflux group with a presumptive cut-off pepsin level based on a previous study [14]. However, the severity of the clinical manifestations did not correlate with the corresponding pepsin levels. Further studies may be necessary using esophageal pH monitoring or esophageal manometry. Our preliminary data should be interpreted with caution and further studies enrolling larger numbers of subjects may, thus, reveal additional microbial patterns. Although we could not give any immunological interpretation and several inherent variabilities between individuals within each cohort, such as the presence of bronchiectasis, should be also considered as potential confounding factors. Despite these limitations, our data supported the impact of reflux on the respiratory microbiota in NTM-PD; this will allow further characterization of reflux in terms of NTM-PD, disease progression, and treatment course. Future research directed towards the role and impact of the respiratory microbiota in NTM-PD progression with the co-existence of reflux is critical to fully elucidate host–microbe interactions and to improve patient care.

## Supporting information

**S1 Fig. Composition of the lower lung microbiome in patients with clinically suspected NTM-PD.** There was no significant difference in the composition of the lower lung microbiome between bacteriologically confirmed NTM-PD (blue dots) and non-NTM-PD group (red dots) (A). Unweighted beta diversity analysis showed the overall bacterial community structure and phylogenetic diversity for bacteriologically confirmed NTM-PD group (blue) and non-NTM-PD group (red)(B).
(TIF)

**S2 Fig. Specific bacterial taxa are associated with the reflux among the patients with bacteriologically confirmed NTM-PD.** The length denotes the effect size for a taxon. $P = 0.05$ for the Kruskal-Wallis test; LDA score >2.5.
(TIF)

**S3 Fig. Predicted functional profiles using PICRUSt2 according to the reflux.** Several functional categories related to aerobactin biosynthesis, nucleic acid/protein/iron metabolism, energy production, or other cellular metabolism were up-regulated in reflux group (A). To the contrary, categories related to antioxidant defense or production of steroid hormones were down-regulated, as such lysine metabolism and steroid biosynthesis in reflux group (B).
(TIF)

**S1 Table. Baseline characteristics of the patients enrolled in this study.**
(DOCX)

## Acknowledgments

We would like to appreciate Hyun Hee Jang for providing technical assistance in the laboratory.

## Author Contributions

**Conceptualization:** Eun Jeong Won, Yong-Soo Kwon.

**Data curation:** Eun Jeong Won.

**Funding acquisition:** Eun Jeong Won, Yong-Soo Kwon.

**Investigation:** Yu Jeong Lee, Moon-Ju Kim, Tae-Jong Kim.

**Methodology:** Yu Jeong Lee, Moon-Ju Kim.

**Project administration:** Eun Jeong Won, Yong-Soo Kwon.

**Resources:** Tae-Jong Kim, Hong-Joon Shin, Tae-Ok Kim.

**Supervision:** Hong-Joon Shin, Tae-Ok Kim.

**Writing – original draft:** Eun Jeong Won, Yong-Soo Kwon.

**Writing – review & editing:** Eun Jeong Won, Tae-Jong Kim, Hong-Joon Shin, Tae-Ok Kim, Yong-Soo Kwon.

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
