## [Decision Letter · Decision Letter 0]

9 Apr 2024

PONE-D-24-06163Lower respiratory tract microbiome in patients with clinically suspected nontuberculous mycobacterial pulmonary disease according to the presence of gastroesophageal refluxPLOS ONE

Dear Dr. Kwon,

Thank you for submitting your manuscript to PLOS ONE. After careful consideration, we feel that it has merit but does not fully meet PLOS ONE’s publication criteria as it currently stands. Therefore, we invite you to submit a revised version of the manuscript that addresses the points raised during the review process.

Please submit your revised manuscript by May 24 2024 11:59PM. If you will need significantly more time to complete your revisions, please reply to this message or contact the journal office at plosone@plos.org. Please include the following items when submitting your revised manuscript:A rebuttal letter that responds to each point raised by the academic editor and reviewer(s). You should upload this letter as a separate file labeled 'Response to Reviewers'.A marked-up copy of your manuscript that highlights changes made to the original version. You should upload this as a separate file labeled 'Revised Manuscript with Track Changes'.An unmarked version of your revised paper without tracked changes. You should upload this as a separate file labeled 'Manuscript'.

We look forward to receiving your revised manuscript.

Kind regards,

Frederick Quinn

Academic Editor

PLOS ONE

Journal Requirements:

https://err.ersjournals.com/content/errev/30/160/200299.full.pdf?

https://www.mdpi.com/2079-6382/10/7/766?

In your revision ensure you cite all your sources (including your own works), and quote or rephrase any duplicated text outside the methods section. Further consideration is dependent on these concerns being addressed.

“This study was supported by grants from the National Research Foundation of Korea (NRF) Grant funded by the Ministry of Education, Science, and Technology (grant no. NRF- 2022R1C1C1002741 (Eun Jeong Won) and 2020R1F1A1076570 (Yong-Soo Kwon)). The funding body played no role in the design of the study and collection, analysis, interpretation of data, and in writing the manuscript.”

5. Please be informed that funding information should not appear in the Acknowledgments section or other areas of your manuscript. We will only publish funding information present in the Funding Statement section of the online submission form. Please remove any funding-related text from the manuscript.

7. We notice that your supplementary figures are included in the manuscript file. Please remove them and upload them with the file type 'Supporting Information'. Please ensure that each Supporting Information file has a legend listed in the manuscript after the references list.

Reviewers' comments:

Reviewer's Responses to Questions

**Comments to the Author**

1. Is the manuscript technically sound, and do the data support the conclusions?

Reviewer #1: Yes

Reviewer #2: Partly

Reviewer #3: Yes

2. Has the statistical analysis been performed appropriately and rigorously? 

Reviewer #1: Yes

Reviewer #2: Yes

Reviewer #3: Yes

3. Have the authors made all data underlying the findings in their manuscript fully available?

Reviewer #1: Yes

Reviewer #2: Yes

Reviewer #3: Yes

4. Is the manuscript presented in an intelligible fashion and written in standard English?

Reviewer #1: Yes

Reviewer #2: Yes

Reviewer #3: Yes

5. Review Comments to the Author

Reviewer #1: This paper illustrated the microbial characteristics in the lower respiratory tract due to reflux in NTM-PD patients. While it may not play a direct role in clinical treatment, I believe it is an extremely important study for understanding the pathophysiology of NTM-PD.

Q1) [Line 89-99 "typical findings of the disease on the chest CT] It would be beneficial to provide additional details on the aspects described in lines 89-99.

Q2) [Line 105 "BAL samples] Were BAL samples adequately collected from all patients (retrieve volume and color)? Were there any samples that did not pass quality control due to low biomass? Was there significant variation in valid read values? How was normalization carried out for read count and gene copy number?

Q3) [Line 157 "24 patients were bacteriologically confirmed NTM-PD] Were these patients confirmed to have NTM through BAL fluid culture?

Q4) [Line 166 Table, NTM-PD with reflux] Shouldn't "NTM-PD with reflux" in Table 1 be accurately labeled as "suspected NTM-PD with reflux"?

Q5) [Line 188-190] It would be advisable to mention in the limitations that Pseudomonas aeruginosa can also be found in conditions of severe lung parenchymal damage (not just reflux). Were there any clinical differences between NTM-PD patients without reflux and those with reflux? (e.g., extent of lung involvement or bacterial load).

Reviewer #2: 16s gene amplicon sequencing to characterize the microbiota of lower respiratory and GI tract from clinical samples.

Patients suspected NTM-PD were selected and included cohorts were NTM-PD was confirmed by bacteriological assay, as well as individuals without confirmation. The samples were further divided between those with and without reflux. Bronchoscopy samples were used for DNA extraction and amplicon sequencing. While no significant difference in overall diversity among patients with suspected NTM-PD, lower diversity was observed in the reflux group. Specific bacterial taxa were found to be associated with efflux.

Despite the potential importance of the questions being addressed, the study is lacking in the level of analysis typically seen for studies assessing microbiota structure. Specific points include:

1. Since 16s gene amplicon sequencing was used to identify bacterial taxa, the microbiome, which includes genes and genomes, was not assessed. References to the microbiome should be changed to “microbiota”.

2. Unfortunately, no LRT samples recovered from bronchoscopy not associated with NTM-PD were analyzed. The lung microbiota should be considered from a greater diversity of samples, even if data from public repositories are analyzed.

3. Beyond bacterial taxa, the authors should identify pathways/functions that correlate with the clinical characteristics. This can be accomplished using standard bioinformatic tools.

4. Fig. 3 claims to show the composition of the lower lung microbiota at the species level. This is placing too much confidence in the results of 16s rRNA gene amplicon sequencing and the figure should be adjusted to show more accurate taxonomic assignments.

5. If reflux is a mechanism for “seeding” the lung with microorganisms from the gut,

6. The limited conclusions from the work do not consider the biology of the bacteria whose differences are noted. A more rigorous assessment of how the bacterial taxa could be relevant. For example Fig. 4 lists strict anaerobic bacteria typically associated with the gut microbiota. What are the implications of these observations?

Reviewer #3: This is a well-written paper and addresses an important topic in NTM-LD. I have a few concerns which are as follows:

1) The introduction needs a hypothesis and summary of results.

2) The authors have chosen all subjects with NTM-LD. This means the cohort must be very diverse. Some patients would have bronchiectasis and others would not. This needs to be discussed and in particular how this must be affecting the microbiome.

3) There are no immunological corrollaries in this study. This is a limitation and therefore must be discussed. The paper is descriptive which is fine but should be discussed as a limitation.

6. PLOS authors have the option to publish the peer review history of their article (what does this mean?). If published, this will include your full peer review and any attached files.

Reviewer #1: No

Reviewer #2: No

Reviewer #3: No

---

## [Author Response · Author response to Decision Letter 0]

10 May 2024

Academic Editor

PLOS ONE

Journal Requirements:

Answer: We have modified a revised manuscript according to the PLOSONE’s style.

https://err.ersjournals.com/content/errev/30/160/200299.full.pdf?

https://www.mdpi.com/2079-6382/10/7/766?

In your revision ensure you cite all your sources (including your own works), and quote or rephrase any duplicated text outside the methods section. Further consideration is dependent on these concerns being addressed.

Answer: We have re-checked a revised manuscript as recommendation.

“This study was supported by grants from the National Research Foundation of Korea (NRF) Grant funded by the Ministry of Education, Science, and Technology (grant no. NRF- 2022R1C1C1002741 (Eun Jeong Won) and 2020R1F1A1076570 (Yong-Soo Kwon)). The funding body played no role in the design of the study and collection, analysis, interpretation of data, and in writing the manuscript.”

Answer: We have added Funding Statement within cover letter as recommendation.

Answer: We have added Supporting information (Table S1). 

5. Please be informed that funding information should not appear in the Acknowledgments section or other areas of your manuscript. We will only publish funding information present in the Funding Statement section of the online submission form. Please remove any funding-related text from the manuscript.

Answer: We have removed Funding information in the Acknowledgments as recommendation.

Answer: We have added ethics statement in the Methods section as recommendation.

7. We notice that your supplementary figures are included in the manuscript file. Please remove them and upload them with the file type 'Supporting Information'. Please ensure that each Supporting Information file has a legend listed in the manuscript after the references list.

Answer: We have added Supporting information as recommendation.

 

Reviewers' comments:

Reviewer's Responses to Questions

Comments to the Author

1. Is the manuscript technically sound, and do the data support the conclusions?

Reviewer #1: Yes

Reviewer #2: Partly

Reviewer #3: Yes

2. Has the statistical analysis been performed appropriately and rigorously?

Reviewer #1: Yes

Reviewer #2: Yes

Reviewer #3: Yes

3. Have the authors made all data underlying the findings in their manuscript fully available?

Reviewer #1: Yes

Reviewer #2: Yes

Reviewer #3: Yes

4. Is the manuscript presented in an intelligible fashion and written in standard English?

Reviewer #1: Yes

Reviewer #2: Yes

Reviewer #3: Yes

5. Review Comments to the Author

Reviewer #1: This paper illustrated the microbial characteristics in the lower respiratory tract due to reflux in NTM-PD patients. While it may not play a direct role in clinical treatment, I believe it is an extremely important study for understanding the pathophysiology of NTM-PD.

Q1) [Line 89-99 "typical findings of the disease on the chest CT] It would be beneficial to provide additional details on the aspects described in lines 89-99.

Answer: We added the typical findings of chest CT as follow (Lines 99-101). “characterized by bilateral multilobar bronchiectasis and small nodules predominantly observed in the right middle lobe and the lingular segment of the left upper lobe”

Q2) [Line 105 "BAL samples] Were BAL samples adequately collected from all patients (retrieve volume and color)? Were there any samples that did not pass quality control due to low biomass? Was there significant variation in valid read values? How was normalization carried out for read count and gene copy number?

Answer: During bronchoscopy, BAL samples were collected by instilling 30 mL of 0.9% NaCl three times into the lung at a subsegmental level, where the lesions appeared to have the most active infection with micronodules and bronchiectasis. The first and third fractions of specimens were sent for a diagnostic microbiological evaluation as part of standard care. Second fraction of specimen was immediately transferred onto ice to lab. Samples were promptly spun at 10,000 g, within 30 minutes of BAL sample collection. The cell pellets in 1 mL were stored at −80°C . DNA was extracted and then was sent to Macrogen for 16S rRNA sequencing. During the two steps of library quality control, 23 samples were excluded because of a low quantity of DNA extracted from specimens. Finally, 47 cases were included for this microbiome study. Mean (SD) value of total reads of each sample was 63,453 (16, 418) reads, and there was no significant variation among 47 samples included in this study. We have described about our procedure in detail in revised manuscript (Lines 106-112; 129). 

Q3) [Line 157 "24 patients were bacteriologically confirmed NTM-PD] Were these patients confirmed to have NTM through BAL fluid culture?

Answer: Yes. All 24 patients tested positive for NTM culture in BAL fluid specimens 

Q4) [Line 166 Table, NTM-PD with reflux] Shouldn't "NTM-PD with reflux" in Table 1 be accurately labeled as "suspected NTM-PD with reflux"?

Answer: We have modified labeling within Table 1.

Q5) [Line 188-190] It would be advisable to mention in the limitations that Pseudomonas aeruginosa can also be found in conditions of severe lung parenchymal damage (not just reflux). Were there any clinical differences between NTM-PD patients without reflux and those with reflux? (e.g., extent of lung involvement or bacterial load).

Answer: Thank you for your valuable comments. We agreed that that Pseudomonas aeruginosa is commonly associated with structural lung diseases such as bronchiectasis, potentially leading to lung parenchymal damage. Therefore, patients with reflux may exhibit more severe lung involvement and a higher prevalence of P. aeruginosa colonization. Our analysis categorized the extent of lung involvement into mild, moderate, and severe based on the number of affected lobes and the presence of bilateral disease or cavities in chest CT scans. Among our patient cohort, 25 individuals (53.2%) exhibited mild disease, while 22 (26.8%) showed moderate disease; no severe cases were recorded. Notably, patients with reflux were more likely to present with moderate disease compared to those without reflux (17 [77.3%] vs. 5 [20.0%], p = 0.001), indicating a greater extent of lung involvement in this subgroup. However, it remains unclear whether this difference in disease severity correlates with alterations in the microbiome, posing a potential limitation to our study. We added it in the Discussion section (Lines 274-285). 

 

Reviewer #2: 16s gene amplicon sequencing to characterize the microbiota of lower respiratory and GI tract from clinical samples. Patients suspected NTM-PD were selected and included cohorts were NTM-PD was confirmed by bacteriological assay, as well as individuals without confirmation. The samples were further divided between those with and without reflux. Bronchoscopy samples were used for DNA extraction and amplicon sequencing. While no significant difference in overall diversity among patients with suspected NTM-PD, lower diversity was observed in the reflux group. Specific bacterial taxa were found to be associated with efflux.

Despite the potential importance of the questions being addressed, the study is lacking in the level of analysis typically seen for studies assessing microbiota structure. Specific points include:

1. Since 16s gene amplicon sequencing was used to identify bacterial taxa, the microbiome, which includes genes and genomes, was not assessed. References to the microbiome should be changed to “microbiota”.

Answer: We have changed references as reviewer’s recommendation.

2. Unfortunately, no LRT samples recovered from bronchoscopy not associated with NTM-PD were analyzed. The lung microbiota should be considered from a greater diversity of samples, even if data from public repositories are analyzed.

Answer: We have added above limitations in revised manuscript as reviewer’s recommendation (Lines 287-288; 297-301).

3. Beyond bacterial taxa, the authors should identify pathways/functions that correlate with the clinical characteristics. This can be accomplished using standard bioinformatic tools.

Answer: We entirely agree the need to further identify the pathways/functions that correlate with the clinical characteristics. Although this provisional study could not add aforementioned data, our data could support the need to large-scale study with tunned scheme for elucidating this notion. We have added above limitations in revised manuscript according to the reviewer’s comments (Lines 297-301).

4. Fig. 3 claims to show the composition of the lower lung microbiota at the species level. This is placing too much confidence in the results of 16s rRNA gene amplicon sequencing and the figure should be adjusted to show more accurate taxonomic assignments.

Answer: The number of listing of taxonomic assignments may appear somewhat arbitrary. We aimed to present this figure in a way that it could infer the composition of the lower lung microbiota between two groups. Therefore, we believe that the top 15 taxonomic assignments provide sufficient insight into their composition patterns in majority. If it is acceptable, we would prefer to keep the current figure unchanged. 

5. If reflux is a mechanism for “seeding” the lung with microorganisms from the gut,

Answer: Thank you for your insightful comment. Reflux has the potential to alter the microbiome by either seeding microbes from the stomach or modifying the respiratory tract environment through the aspiration of gastric contents. However, we could not explain the precise mechanisms underlying these microbiome changes in patients with reflux in this study (Lines 239-254).

6. The limited conclusions from the work do not consider the biology of the bacteria whose differences are noted. A more rigorous assessment of how the bacterial taxa could be relevant. For example Fig. 4 lists strict anaerobic bacteria typically associated with the gut microbiota. What are the implications of these observations?

Answer: We have more discussed about the biology of the bacteria whose differences were noted, such as anaerobic bacteria associated with the gut microbiota, in revised manuscript (Lines 239-254).

 

Reviewer #3: This is a well-written paper and addresses an important topic in NTM-LD. I have a few concerns which are as follows:

1) The introduction needs a hypothesis and summary of results.

Answer: We have already delineated the hypothesis of this study concerning the variances in the microbiome's composition in the presence of GERD (Lines 69-91). 

2) The authors have chosen all subjects with NTM-LD. This means the cohort must be very diverse. Some patients would have bronchiectasis and others would not. This needs to be discussed and in particular how this must be affecting the microbiome.

Answer: We have added above limitations in revised manuscript as reviewer’s recommendation (Lines 286-303).

3) There are no immunological corrollaries in this study. This is a limitation and therefore must be discussed. The paper is descriptive which is fine but should be discussed as a limitation.

Answer: We have added above limitations in revised manuscript as reviewer’s recommendation (Lines 299-303).

6. PLOS authors have the option to publish the peer review history of their article (what does this mean?). If published, this will include your full peer review and any attached files.

Do you want your identity to be public for this peer review? For information about this choice, including consent withdrawal, please see our Privacy Policy.

Reviewer #1: No

Reviewer #2: No

Reviewer #3: No

While revising your submission, please upload your fig

---

## [Decision Letter · Decision Letter 1]

14 Jun 2024

PONE-D-24-06163R1Lower respiratory tract microbiome in patients with clinically suspected nontuberculous mycobacterial pulmonary disease according to the presence of gastroesophageal refluxPLOS ONE

Dear Dr. Kwon,

Thank you for submitting your manuscript to PLOS ONE. After careful consideration, we feel that it has merit but does not fully meet PLOS ONE’s publication criteria as it currently stands. Therefore, we invite you to submit a revised version of the manuscript that addresses the points raised during the review process.

Please submit your revised manuscript by Jul 29 2024 11:59PM. If you will need significantly more time to complete your revisions, please reply to this message or contact the journal office at plosone@plos.org. Please include the following items when submitting your revised manuscript:A rebuttal letter that responds to each point raised by the academic editor and reviewer(s). You should upload this letter as a separate file labeled 'Response to Reviewers'.A marked-up copy of your manuscript that highlights changes made to the original version. You should upload this as a separate file labeled 'Revised Manuscript with Track Changes'.An unmarked version of your revised paper without tracked changes. You should upload this as a separate file labeled 'Manuscript'.

We look forward to receiving your revised manuscript.

Kind regards,

Frederick Quinn

Academic Editor

PLOS ONE

Reviewers' comments:

Reviewer's Responses to Questions

**Comments to the Author**

1. If the authors have adequately addressed your comments raised in a previous round of review and you feel that this manuscript is now acceptable for publication, you may indicate that here to bypass the “Comments to the Author” section, enter your conflict of interest statement in the “Confidential to Editor” section, and submit your "Accept" recommendation.

Reviewer #1: All comments have been addressed

Reviewer #2: (No Response)

2. Is the manuscript technically sound, and do the data support the conclusions?

Reviewer #1: Yes

Reviewer #2: Partly

3. Has the statistical analysis been performed appropriately and rigorously? 

Reviewer #1: Yes

Reviewer #2: Yes

4. Have the authors made all data underlying the findings in their manuscript fully available?

Reviewer #1: Yes

Reviewer #2: Yes

5. Is the manuscript presented in an intelligible fashion and written in standard English?

Reviewer #1: Yes

Reviewer #2: Yes

6. Review Comments to the Author

**Reviewer #1: **The authors of the paper adequately answered the questions. Thank you for providing the opportunity to read such a good paper.

**Reviewer #2: **While the revised version is improved, the omission of standard bioinformatics analysis to predict gene function rather than only catalog bacterial taxonomy continues to limit the impact of the manuscript.

Comments

Reviewer number two suggested a more comprehensive analysis of the data to identify features of the samples that are more informative than bacterial taxa alone. For example, PICRUSt (now PICRUSt2, DOI: 10.1038/s41587-020-0548-6), which predicts functional potential of a bacterial community based on marker gene sequencing profiles, is a standard tool for analyzing 16s amplicon sequencing data. The continued lack of marker gene analysis leaves the reader lacking potentially useful information that can better inform future studies.

The following misstatements should be corrected:

1. 16S ribosomal RNA sequencing was not done, rather 16S ribosomal RNA gene amplicon sequencing was performed.

2. 16s amplicon sequencing is used to interrogate the microbiota, not the microbiome, as specific genes and genome are not being characterized. This was noted by original reviewer #2 but was not changed in the title, keywords and abstract.

3. The abstract states that: “These observations confirm that the lower respiratory microbiome is reliably altered by reflux but not in NTM-PD”. The word “reliably” is awkward and should be replaced with “consistently”.

7. PLOS authors have the option to publish the peer review history of their article (what does this mean?). If published, this will include your full peer review and any attached files.

Reviewer #1: No

Reviewer #2: No

---

## [Author Response · Author response to Decision Letter 1]

19 Jul 2024

Reviewer number two suggested a more comprehensive analysis of the data to identify features of the samples that are more informative than bacterial taxa alone. For example, PICRUSt (now PICRUSt2, DOI: 10.1038/s41587-020-0548-6), which predicts functional potential of a bacterial community based on marker gene sequencing profiles, is a standard tool for analyzing 16s amplicon sequencing data. The continued lack of marker gene analysis leaves the reader lacking potentially useful information that can better inform future studies.

Answer: We have added PICRUSt data as S3 Figure and we have described functional data with discussion in revised manuscript accordingly (Lines 151-152; 217-224; 298-304). We have added more references and rearranged them accordingly.

The following misstatements should be corrected:

1. 16S ribosomal RNA sequencing was not done, rather 16S ribosomal RNA gene amplicon sequencing was performed.

Answer: We have corrected the word throughout manuscript as recommendation.

2. 16s amplicon sequencing is used to interrogate the microbiota, not the microbiome, as specific genes and genome are not being characterized. This was noted by original reviewer #2 but was not changed in the title, keywords and abstract.

Answer: We have revised the word throughout manuscript as recommendation.

3. The abstract states that: “These observations confirm that the lower respiratory microbiome is reliably altered by reflux but not in NTM-PD”. The word “reliably” is awkward and should be replaced with “consistently”.

Answer: We have corrected the word as recommendation (Line 59).

---

## [Decision Letter · Decision Letter 2]

13 Aug 2024

Lower respiratory tract microbiota in patients with clinically suspected nontuberculous mycobacterial pulmonary disease according to the presence of gastroesophageal reflux

PONE-D-24-06163R2

Dear Dr. Kwon,

We’re pleased to inform you that your manuscript has been judged scientifically suitable for publication and will be formally accepted for publication once it meets all outstanding technical requirements.

Kind regards,

Frederick Quinn

Academic Editor

PLOS ONE

Additional Editor Comments (optional):

Reviewers' comments:

Reviewer's Responses to Questions

**Comments to the Author**

1. If the authors have adequately addressed your comments raised in a previous round of review and you feel that this manuscript is now acceptable for publication, you may indicate that here to bypass the “Comments to the Author” section, enter your conflict of interest statement in the “Confidential to Editor” section, and submit your "Accept" recommendation.

Reviewer #1: All comments have been addressed

Reviewer #2: All comments have been addressed

2. Is the manuscript technically sound, and do the data support the conclusions?

Reviewer #1: Yes

Reviewer #2: Yes

3. Has the statistical analysis been performed appropriately and rigorously? 

Reviewer #1: Yes

Reviewer #2: Yes

4. Have the authors made all data underlying the findings in their manuscript fully available?

Reviewer #1: Yes

Reviewer #2: Yes

5. Is the manuscript presented in an intelligible fashion and written in standard English?

Reviewer #1: Yes

Reviewer #2: Yes

6. Review Comments to the Author

Reviewer #1: The authors responded well to the reviewers' questions. There are no additional questions. This paper is expected to make significant contributions to the study of the microbiome in the field of NTM pulmonary diseases.

Reviewer #2: This is a second revision of a manuscript that describes the use of 16s taxonomic profiling to characterize the lung microbiota from individuals with different NTM-PD status. In particular, the study seeks to understand the potential impact of GE reflux. The manuscript reports the identification of bacterial taxa that are associated with the patients experiencing reflux.

The authors have responded appropriately to the previous reviews of the manuscript.

7. PLOS authors have the option to publish the peer review history of their article (what does this mean?). If published, this will include your full peer review and any attached files.

Reviewer #1: No

Reviewer #2: No

---

## [Editor Report · Acceptance letter]

19 Aug 2024

PONE-D-24-06163R2 

PLOS ONE

Dear Dr. Kwon, 

I'm pleased to inform you that your manuscript has been deemed suitable for publication in PLOS ONE. Congratulations! Your manuscript is now being handed over to our production team.

Kind regards, 

on behalf of

Dr. Frederick Quinn 

Academic Editor

PLOS ONE